# The Use of EU Territorial Cooperation Funds for the Sustainable Development of National and Ethnic Minorities in the Baltic Sea Region

**Tomasz Studzieniecki** [1,*] and **Beata Meyer** [2]

1   Department of Management and Economics, Gdynia Maritime University, 81-225 Gdynia, Poland
2   Institute of Spatial Economy and Socio-Economic Geography, University of Szczecin, 71-017 Szczecin, Poland; beata.meyer@usz.edu.pl
*   Correspondence: t.studzieniecki@wznj.umg.edu.pl; Tel.: +48-604-99-66-36

**Abstract:** The integration processes taking place in the Baltic Sea Region (BSR) are contributing to the sustainable development of this unique territory. Intensive cooperation financed from EU funds under the cohesion policy and the neighborhood policy have provided an opportunity for the development of ethnic and national minorities, who are important but still marginalized stakeholders. The theoretical aim of the article is to identify the attributes of national and ethnic minorities and to indicate key determinants of their sustainable development. The authors seek to answer whether territorial cooperation in the BSR contributes to the development of national and ethnic minorities, and if so, how. In the article, concepts and definitions related to the term "minority" are verified, classifications are developed and a model of sustainable development of ethnic and national minorities is built. Official statistical data are used to present the national and ethnic structure of the BSR countries. Then, 22 national minorities and 17 ethnic minorities are identified and described. Quantitative and qualitative research was carried out on 126 territorial cooperation projects supporting the development of national and ethnic minorities totaling EUR 85.25 million in value and implemented within 38 BSR programs in 2000–2020. The Sami minority were the greatest beneficiaries of the cooperation. Territorial cooperation projects have been shown to contribute to the preservation of cultural heritage and the development of education, social support and political empowerment. Territorial cooperation is a powerful instrument of sustainable development. Unfortunately, it still contributes only moderately to the development of national and ethnic minorities. There is a need to strengthen this issue in future programs of the cohesion policy and the neighborhood policy and to develop systemic solutions enabling national and ethnic minorities to participate more actively in the implementation of territorial cooperation projects.

**Keywords:** sustainable development; ethnic; national; minority; territorial cooperation; Baltic

## 1. Introduction

The Baltic Sea Region is a unique transnational area characterized by complex integration and disintegration processes. It is also a colorful cultural, linguistic and religious ethnic mosaic. By the end of the 20th century, many conflicts arose in the Baltic Sea Region between rival nations, states and international alliances. However, at the beginning of the 21st century, thanks to geopolitical stabilization, there was an unprecedented intensification of territorial cooperation. Unfortunately, this cooperation was severely limited with the outbreak of the COVID-19 pandemic and the resurgence of Russian imperialism.

It is noteworthy that for several decades key decision-makers and leaders of international cooperation in the BSR have been systematically implementing ambitious sustainable development goals. This situation reflects global tendencies to reduce environmental threats by improving the theory and practice of sustainability. The paradigm of sustainable development (SD) emerges in many theoretical concepts and practical activities concerning

the development of social minorities, including national and ethnic minorities [1–5]. The idea of sustainability unites diverse social groups, including politicians dependent on the electorate, business-driven entrepreneurs and activists fighting for social justice [6]. SD is a broad and ubiquitous concept [7] that has entered the jargon of decision makers, a catchphrase for international initiatives, a theme of conferences and a slogan used in regulations or declarations [8–10].

Observing the processes of Baltic integration and actively participating in numerous initiatives for the development of the Baltic Sea Region, the authors of this article noticed that national and ethnic minorities rarely appear among the stakeholders and beneficiaries of territorial cooperation. For this reason, the authors decided that this issue was so important that it should be thoroughly analyzed. In the first stage, literature studies were conducted on the determinants of sustainable development in connection with the functioning and development of national and ethnic minorities. The result of this research was the creation of a model of sustainable development of national and ethnic minorities. Then, on the basis of official statistical data, national and ethnic minorities were identified. It has been noticed that the definitions, regulations and policies of ethnic minorities in the BSR are extremely diverse. In the next stage, the project database (keep.eu) was used, and a quantitative and qualitative analysis of all projects supporting the development of national and ethnic minorities was carried out. Research has shown that the instruments of territorial cooperation moderately support the sustainable development of national and ethnic minorities.

It is logical that this situation requires improvement, as the issue of including social minorities in the process of multilevel cooperation is particularly important in the context of achieving the goals of sustainable development, preventing social exclusion and reducing the danger of ethnic and national conflicts.

## 2. The Key Determinants of the Identification and Sustainable Development of Ethnic and National Minorities

Three epochs can be distinguished in the process of defining sustainable development. During the first epoch, the boundaries of development and environmental requirements were recognized. The Club of Rome warned of the negative consequences of economic development [11], and the Stockholm conference developed recommendations for the development of the human environment [12]. The second period began with the Stockholm conference, which refined the concept of development without destruction [13]. The third, so called Post-Brundtland period is still current [14]. It assumes a holistic approach to sustainable development that includes three key elements—the environment, economy and society. The relationships between the three sustainability components (Figure 1) have been visualized as three intersecting circles, concentric circles and pillars [15].

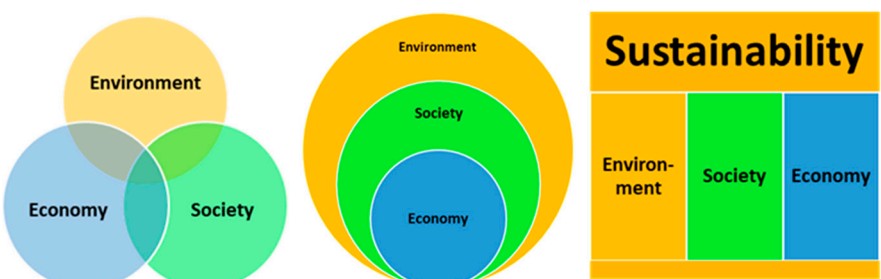

**Figure 1.** Visualization of relations between the three elements of sustainable development. Source: own study based on [15].

In relation to the three elements of sustainable development, they have also been referred to as aspects [16], perspectives [17], pillars [18] and dimensions [19].

Analysing the literature on sustainable development, Klarin [20] and Borowy [21] noticed that one of the most-cited definitions is the UN definition from 1987 [22], according

to which sustainable development meets the needs of the present without compromising the ability of future generations to meet their own needs. In understanding the essence of the definition, it is helpful to conceptualize three concepts of development, needs and future generations (Figure 2).

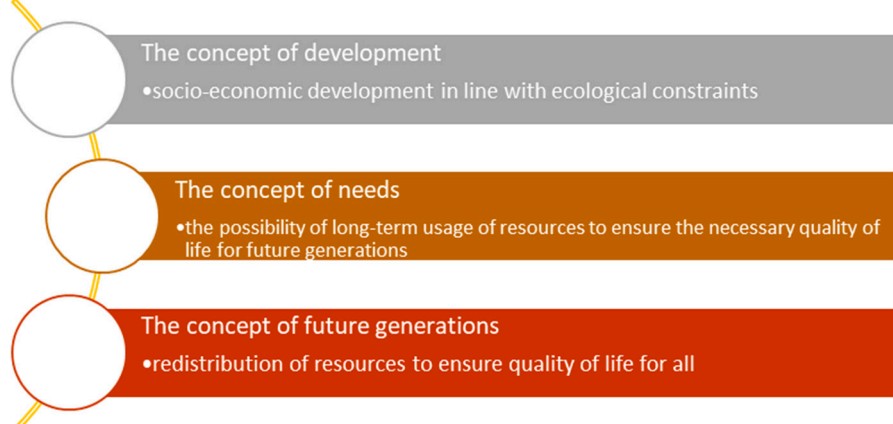

**Figure 2.** Conceptualization of the key ideas of sustainable development. Source: own study based on [20].

The "Earth Summit" held in Rio de Janeiro in 1992 defined principles of sustainable development that in harmony with nature should contribute to sustaining human life in the future. It also proposed a vision for a method of new economic and social development around the world [23]. The conference listed fundamental principles on which countries should base their socio-economic development. It provided a comprehensive framework for the management of sustainability issues, defining the rights and obligations of nations [24]. According to the *Rio Declaration*, "Human beings are at the center of concerns for sustainable development. They are entitled to a healthy and productive life in harmony with nature".

At the same time, the *Global Agenda 21 Action Programme* indicated actions necessary to achieve sustainable development and improve quality of life [25]. The document contains a comprehensive and global programme of desirable development and environmental actions reflecting the global consensus and political commitment to development and cooperation in the field of environment in the 21st century [26]. Attention was drawn to the problem of sustainability in terms of population growth, poverty and health, as well as urbanization in the context of the environment and the health of citizens.

The UN Millennium Summit held in New York in 2002 created the Millennium Declaration, which requested its signatories to act for peace, security, disarmament, development and the eradication of poverty. The Millennium Development Goals were specified in the form of eight commitments that humanity should meet in order to be able to effectively address the challenges of the 21st century. The provisions of the UN Earth Summit held in Johannesburg in the same year have been assessed as vague and having repeated prior commitments to *Agenda 21* and the *UN Millennium Declaration* [27]. The Earth Summit highlighted the importance of civil society in implementing results and promoting partnership initiatives.

In 2012, the Rio+20 conference took place (following on from the 1992 conference) and aimed to renew political commitments to sustainable development, assess progress and identify barriers to the implementation of key commitments. One of important considerations was social sustainability [28]. The document recognized the fight against poverty as its most important objective. The UN Summit held in New York in 2015 adopted the *2030 Agenda*, which contained a set of 17 Sustainable Development Goals, 169 specific tasks and over 230 indicators. The document indicated the need for economic and social development ensuring the well-being of citizens while respecting the natural environment [29].

In the context of the documents presented above, it can be stated that the stakeholders and beneficiaries of sustainable development comprise all social groups, regardless of gender,

age, religion, race, language or culture. Sustainable development in social terms must take into account the interests of social minorities, including national and ethnic minorities.

Problems related to the functioning and development of "minority groups" [30,31] are richly reflected in the scientific literature [32–34]. This issue is analyzed by representatives of many scientific disciplines, including sociologists, political scientists, historians, geographers, economists and lawyers. The multitude of approaches, concepts, definitions and typologies of minority groups [35] make it difficult to self-identify [36] and identify [37] these communities and to find out the determinants of their development [38].

According to Pierre George, the term "minority" means groups of people who are inferior in number and simultaneously marginal politically, socially and economically, as well as culturally [39]. Fortman emphasizes that there should be nothing normative in the notion of a minority. According to him, it is just a residual category in respect to the idea of majority rule [40]. However, research shows that in many cases the term "minorities" has negative connotations related to fear of a foreign culture, deterioration of the labor market and crime [41].

Among the many definitions of minorities [42], the definition created for the needs of the United Nations [43] has been widely used in theory and practice. According to this definition, a minority is a group numerically inferior to the rest of the population of a state, in a non-dominant position, whose members—being nationals of the state—possess ethnic, religious or linguistic characteristics that differ from the rest of the population and show, if only implicitly, a sense of solidarity directed towards preserving their culture, traditions, religion or language. Jackson-Preece notes [44] that while ethnic, religious or linguistic characteristics can be easily verified, the identification of a "sense of solidarity" is subjective.

In the context of UN regulations on the protection of minority rights [45], four basic minority groups are distinguished: national, ethnic, religious and linguistic. Distinguishing between a national and an ethnic minority is a difficult and controversial task [46]. This issue has a political aspect and it is regulated by state authorities [47], so the definitions and classifications are arbitrary. Koter suggests [48] that minorities should be divided into seven groups belonging to two collections (Table 1).

**Table 1.** Genetic classification of minorities.

| No. | Minority Group | Genesis |
| --- | --- | --- |
| 1. | "Aborigines", "tribals" and other primary groups of archaic origin | Autochthons |
| 2. | Other ethnic groups and nationalities of very old origin | Autochthons |
| 3. | Invaders and conquerors | Newcomers |
| 4. | Settlers (settled by conquerors or by their own rulers or possessors) | Newcomers |
| 5. | Displaced persons | Newcomers |
| 6. | Refugees | Newcomers |
| 7. | Gastarbeiters | Newcomers |

Source: own study based on [48].

The identification of criteria such as country of birth or citizenship become useful when focusing on social problems related to the situation of recently immigrated groups of the population [49]. As the equivalent of a foreign background, a differentiation according to ethnic or cultural background is often used [50]. In some countries (e.g., Germany) that have historically developed a national group membership driven by the idea of heritage, immigrants are often perceived as foreigners even after their naturalization [51]. Following the principles of political correctness, data on racial and ethnic origin are operationalized as migration background markers and other proxies (Table 2). The broadest definition of a migration background is the definition of the micro-census, while other specific statistics use diverging definitions, which creates problems regarding comparability [52].

**Table 2.** Criteria for the division of the residents of Germany into Germans and foreigners.

| No. | Criterion | Germans | Foreigners |
|---|---|---|---|
| 1. | without migration background | "native" Germans | - |
| 2. | with migration background | born into a family which came to Germany after 1955 | holding foreign nationality |
| 3. | with migration experience | Germans who migrated themselves (e.g., Germans from the former Soviet Union) | born outside Germany |
| 4. | without migration experience | born in Germany | born in Germany |

Source: [52].

Completely different categories and criteria for the division of minorities are used in Poland [53], where three categories of minority groups have been officially distinguished, namely national minorities, ethnic minorities and minorities using a regional language.

The definitions of the groups mentioned in Table 3 are specified in Polish law. The interpretation of the regulations and the identification of minority groups are carried out by the government administration. The guidelines of the Council of Europe [54] apply when identifying a group using a regional language.

**Table 3.** Criteria for identifying minority groups in Poland.

| No | Criterion | National Minority | Ethnic Minority | Regional Language Minority |
|---|---|---|---|---|
| 1. | The group identifies with a nation that elsewhere has its own state. | + | | |
| 2. | The group does not identify with a nation that elsewhere has its own state. | | + | |
| 3. | The group strives to preserve its language, culture or tradition. | + | + | |
| 4. | The group differs significantly from other citizens in terms of language, culture or tradition. | + | + | |
| 5. | The group is aware of its own historical national or ethnic heritage and aims to express and protect it. | + | + | |
| 6. | The ancestors of the group lived in the present territory of the country for at least 100 years. | + | + | |
| 7. | The group is less numerous than the rest of the country's population. | + | + | |
| 8. | The group uses a language other than the official language of that country; this includes neither dialects of the state's official language nor the languages of migrants. | | | + |

Source: own study based on the [53].

Numerous studies prove that the public distinguish little between national minorities, ethnic minorities and those using a regional language [55,56]. In everyday language, all of these terms are often used interchangeably. In such a situation, it is advisable to use the term "ethno-national minorities" [57], which is more conciliatory and pragmatic. A national ethnic minority can be understood as a social group that is aware of its individuality or that coheres through shared experiences [58]. Such a minority can be identified by race, religion or national origin, or by combining these categories as a whole [59,60]. An important criterion for classifying minorities is the question of territory.

Minorities can be divided into those that have and do not have their own territory. The following minorities can be distinguished by location [61]:

1.    Occurring in one country (e.g., Sorbs in Germany);

2. Occurring in several countries (e.g., the Sami people in Scandinavia);
3. Occurring in many countries (e.g., Roma all over the world).

Minorities can be concentrated in one place or several places, or dispersed throughout the territory of a given country [62]. It becomes useful to distinguish the category of "minorities in border areas", the development of which is often stimulated by cross-border cooperation [63–65]. When making a typology of national and ethnic minorities, three options and seven types of minorities can be distinguished in the context of the state and its borders (Figure 3). Option 1 is the situation in which a minority or minorities are outside of the country's borders. The second option is the situation in which a given minority exists only in the territory of a given state. It can be located in one place (2a), in several places (2b) or spread over the territory of a given country. The third option is the situation in which a given minority occurs both within the territory of a given country and abroad. Such a minority may be located only on the border of a given country and a neighboring country (3a), in several countries (3b) or scattered around the world (3c).

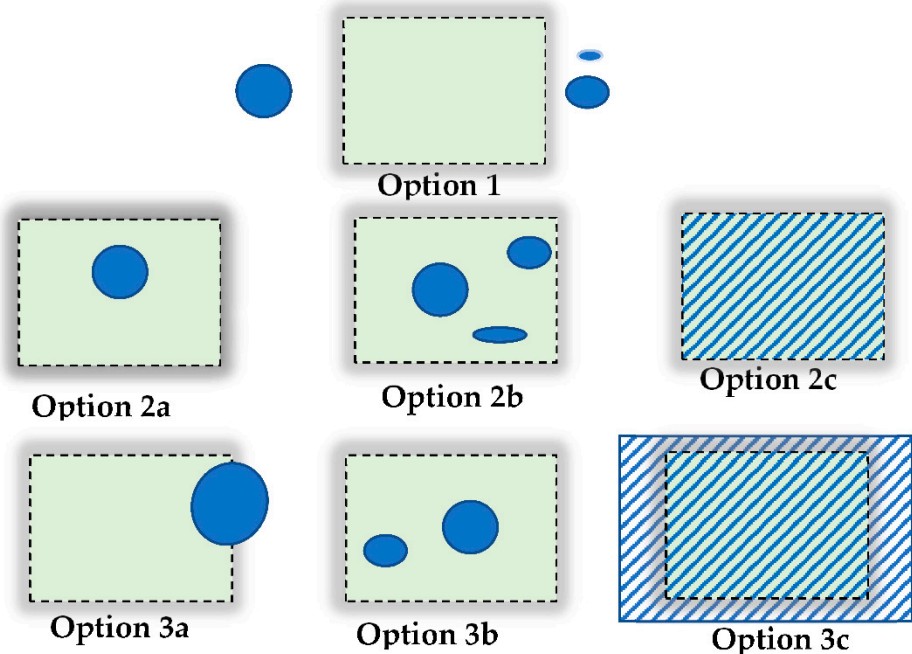

**Figure 3.** Classification of ethno-national minorities. Source: own study.

Concepts regarding the approach to and treatment of ethnic minorities are very diverse [66–69]. It should be an ethical imperative [70] that the approach to minorities not involve discrimination against them by politicians [71] or society [72] (including by other minorities) [73]. It is important to take measures to prevent exclusion and to encourage inclusion [74]. There are many forms of minority exclusion [75], but they can be reduced to four groups (Figure 4) [76]. The views of theoreticians and decision-makers regarding the provision of autonomy to minorities are diverse. Wright believes [77] that autonomy for minority groups is an appropriate mechanism through which a state's obligation to afford a right of self-determination to all its peoples may be fulfilled.

Smith distinguishes two types of autonomy—territorial and non-territorial [78]. In turn, Benedikter adds one more category—local (administrative) autonomy [79]. According to Lapidoth [80], territorial autonomy is an agreement that enables a given group that differs from most of society but is the majority in a given region to be equipped with the means to express its individuality. In turn, in non-territorial forms of autonomy, the beneficiary of self-government is neither the individual nor a specified area but the ethnic group itself, understood as a corporate unit [81].

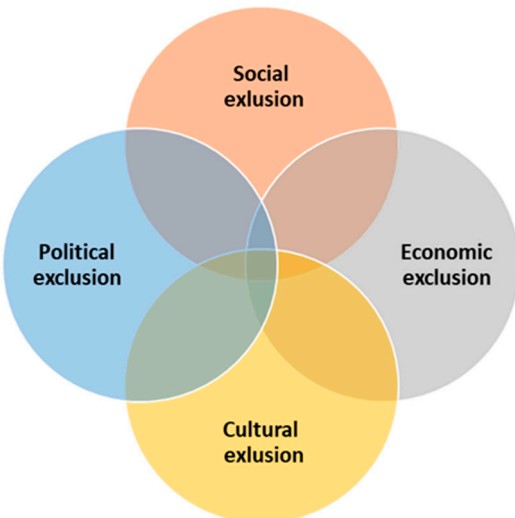

**Figure 4.** Main forms of minority exclusion. Source: own study based on [76].

An extremely important factor in the development of minorities is the cooperation of minorities both with the (majority) society [82] and with other minorities, including territorial cooperation, the main stakeholders of which are sub-state entities [83]. The cooperation should lead to the well-being [84] of ethno-national minorities. Experts from The International Bank for Reconstruction and Development indicate internal and external factors that determine the well-being of minorities [75] (Table 4).

**Table 4.** Driving forces of development of selected ethno-national minority groups.

| Minority Well-Being | |
|---|---|
| **Direct Factors** | **Indirect Factors** |
| 1. Political connectedness | 1. Labor market |
| 2. Gender roles and power | 2. Physical and economic connectivity |
| 3. Mass organisation membership | 3. Market linkages |
| 4. Inputs, subsidies | 4. Vulnerabilities and shocks |
| 5. Credit | 5. Local governance |
| 6. Remittances | 6. Traditional institutions |
| 7. Health conditions | 7. Culture, spiritual beliefs, religion |
| 8. Knowledge, skills | 8. Misperceptions and stigma |
| 9. Labor | 9. Donor-supported projects |
| 10. Landholding | 10. Gov't policies and programs |
| 11. Social network and kinship | 11. Basic public services |

Source: own study based on [75].

In connection with the principles of sustainable development, it is becoming necessary to provide national and ethnic minorities with basic values and rights that affect their quality of life, including:

1. Access to services and social benefits [85];
2. Participation in political decision-making [86];
3. Preservation of cultural and natural heritage [87].

The issues of the development of national and ethnic minorities in the *2030 Agenda* have been treated quite marginally. The document emphasizes the need to strengthen vulnerable members of society, including indigenous peoples, refugees and internally displaced persons and migrants. These social groups "should have access to life-long learning opportunities that help them acquire the knowledge and skills needed to exploit opportunities and to participate fully in society". The document emphasizes the need for:

1. Monitoring the condition of minorities, including race, ethnicity and migratory status;

2.  Implementing anti-discrimination measures;
3.  Social, economic and political inclusion;
4.  Better access to education;
5.  Support for agricultural activities "through secure and equal access to land, other productive resources and inputs, knowledge, financial services, markets and opportunities for value addition and non-farm employment".

In the context of the sustainable development of ethnic minorities, three main goals of development can be identified, i.e., equality, prosperity and vitality, and 17 specific goals of sustainable development indicated in the *2030 Agenda* can be assigned to them (Figure 5). In a holistic approach to the problem of minority development, it is helpful to use the concept of multilevel cooperation [88], which allows for the identification of overlapping levels relevant to a given phenomenon [89]. In vertical terms, three levels of cooperation can be distinguished: subnational (local, regional), national (state) and transnational (supranational) [90]. On the other hand, in a horizontal approach, political, legal, economic and social factors occurring at all vertical levels can be considered significant external factors of the development of national and ethnic minorities.

| Equality | Prosperity | Vitality |
|---|---|---|
| • No poverty<br>• Zero Hunger<br>• Good health and well-being<br>• Quality Eduation<br>• Gender Equality | • Clean water and sanation<br>• Affordable and clean energy<br>• Decent work and economic growth<br>• Industry innovation and infrastructure<br>• Reduced inequalities<br>• Sustainable cities and communities | • Responsible production and consumption<br>• Climate Action<br>• Life below water<br>• Life on land<br>• Peace, justice and strong institutions<br>• Partnerships for the goals |

**Figure 5.** The main and specific goals of the sustainable development of national and ethnic minorities. Source: own study.

The identification of levels and external and internal factors makes it possible to develop a model of the development of national and ethnic minorities (Figure 6).

It is assumed that the achievement of the basic goals of sustainable development is determined by internal and external factors. Internal factors depend entirely on the social minorities themselves. They are related to self-identification, self-organization and determination to act, as well as the socio-economic potential enabling the achievement of the set goals. In turn, external factors are related to the environment covering 3 levels:

1.  Subnational (e.g., city, administrative region);
2.  National (state);
3.  Transnational (e.g., Euroregion, European Union).

The most important external factors are: political factors (e.g., the ethnic policy of the state), legal factors (local, regional, state and EU regulations), economic factors (e.g., the state budget, the amount of international funds) and social factors (e.g., the education system, social welfare system, society's attitude towards social minorities).

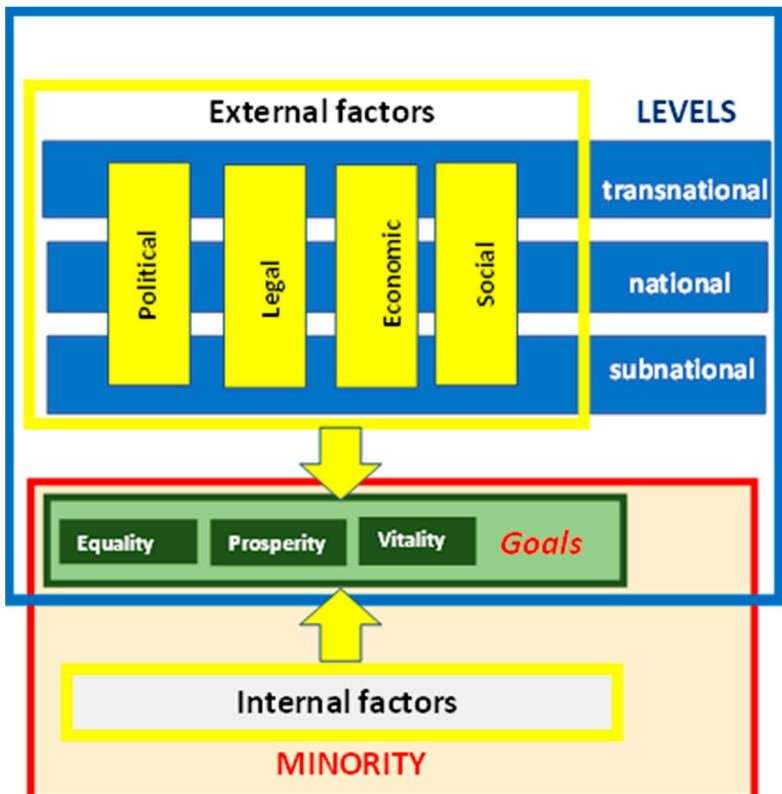

**Figure 6.** Model of the sustainable development of national and ethnic minorities. Source: own study.

## 3. The Genesis and Historical Development of Nations and States of the Baltic Sea Region

The term "Baltic Sea Region" is difficult to define. It can be assumed that these are areas lying on the Baltic Sea, historically and economically linked with this water body [91]. There are countries such as Sweden, Denmark, Finland, Latvia and Estonia whose histories are almost entirely related to the Baltic region. The region also includes countries such as Poland, Germany, Russia and Lithuania in the historical sense (the Grand Duchy of Lithuania), which were intermittently associated with the Baltic Sea and were for a long time turned away from the sea that was closely related to its history. The first traces of settlement date back nearly ten thousand years. Most likely, the ancestors of today's Sami people lived here. Then there were tribes that were Finno-Ugric (today's Finns and Estonians), Balt (Latvians, Lithuanians, Prussians), Germanic (Germans, Danes, Swedes) and Slavic (Poles, Russians). The Germanic Norman peoples inhabiting the territory of Scandinavia in the 7th–9th centuries gained notoriety in Europe for their numerous conquests [92].

It is difficult to say which of the Baltic states was formed first. Representatives of each nation could argue that their country qualifies. It is important that in the 10th century, countries such as Denmark, Sweden, Poland, Germany and Ruthenia appeared on the map of Europe, but in the early Middle Ages these last three countries only sporadically rubbed against the Baltic Sea [93]. Christianity spread in this area, and in the 10th century the above-mentioned countries were already Christian (Denmark AD 960, Poland AD 966, Ruthenia AD 988), while the Finno-Ugric and Baltic peoples continued to cultivate their tribal religions, and it was not until the 13th and 14th centuries that Christianity became established in this area.

Significant changes took place in the 12th and 13th centuries, when the territories of the Polabian Slavs were conquered by German magnates, Sweden made Finland a dependent state and Lubeck (a thriving city on the Baltic Sea) developed and became the leader of the union of cities. In the 13th century, on the shores of the Baltic Sea, knightly orders appeared, including the Teutonic Order in Prussia and the Knights of the Sword in Livonia (today's Latvia and Estonia), giving rise to German settlement in these areas. For the time being,

part of Estonia came under Danish rule. The Baltic Prussians were exterminated by the Teutonic Knights and disappeared.

At the end of the 14th century, as a result of dynastic unions, new political configurations were created and Lithuania joined Poland in 1385. In turn, in 1397 in Kalmar, the three Scandinavian states of Denmark, Sweden and Norway formed a union, which unlike the Polish–Lithuanian one, survived intermittently until 1523. Then, Denmark remained united with Norway, while Sweden and Finland formed a separate entity.

The Reformation influenced the political development of the Baltic Sea Region. Denmark, Sweden, Finland, Norway and Brandenburg became Lutheran. Knightly orders were also secularized. The 17th century was a period of triumphs for Sweden, which tried to transform the Baltic Sea into an internal Swedish one. After the victories over the Habsburgs, Poland, Denmark and Russia, the Swedish kings ruled not only Sweden and Finland, but also Ingria on the Neva, Livonia up to Riga, West Pomerania and Skåne, which was taken back from Denmark in 1658.

The Northern War of the beginning of the 18th century brought an end to Swedish domination. It was then that the victorious Russia appeared on the Baltic Sea, occupying Livonia and the mouth of the Neva, where Tsar Peter I built a new capital of the Roman Empire—St Petersburg. Prussia was strengthened by joining Western Pomerania. The partitions of Poland brought Russia further acquisitions on the Baltic Sea, namely Courland and Lithuania, while Prussia took Pomerania along with Gdańsk and Elbląg. Subsequent changes took place during the Napoleonic Wars. Russia took Finland from Sweden in 1809, creating an autonomous Grand Duchy of Finland, and Denmark lost Norway, which was joined by the Union with Sweden under the Congress of Vienna in 1815. The 19th century was a period of domination over the Baltic Sea by Russia and Prussia. In 1871, a strong united Germany emerged in the place of Prussia. The 19th century was a period of political stabilization on the Baltic Sea.

Significant border changes took place in the 20th century. In 1905, Norway peacefully seceded from Sweden. As a result of the revolution and civil war in Russia and the defeat of Germany in World War I, many Baltic nations gained independence. Finland appeared on the map of Europe as early as 1917, followed by Lithuania and Latvia and Estonia. Poland gained a little access to the Baltic Sea.

Under the Soviet–German Ribbentrop–Molotov treaty of 1940, the Baltic republics of Lithuania were forcibly attached to the USSR. Finland lost Karelia after the war with the Soviet Union in 1939–40 but retained its independence. Poland was under German and Soviet occupation. The new Yalta–Potsdam order divided the Baltic Sea. Many of the states found themselves in the Soviet sphere of influence. The Baltic states remained under Soviet occupation. Konigsberg and the surrounding areas were attached to the USSR as the Kaliningrad Oblast. Poland gained wide access to the Baltic Sea but remained in the Soviet sphere of influence, similarly to the German Democratic Republic established in 1949 from the Soviet occupation zone. Dependent on the USSR was capitalist Finland. Sweden remained neutral, and Denmark, Norway and Germany became NATO members. The democratic Scandinavian countries and western Germany, skilfully taking advantage of the economic situation, have become some of the richest countries in the world.

As a result of World War II, the German population left Latvia and Estonia. They were also displaced from Polish territories and the Soviet Kaliningrad Oblast. The ethnic composition of these areas changed. Russians appeared in Konigsberg and the Baltic republics, and Poles in Szczecin. The Germans ceased to dominate the southern Baltic Sea. In Germany and the Scandinavian countries, they began to settle as Gastarbeiters from distant countries, from Muslim and Arab countries. Africans, Indians, Arabs and Turks are already a permanent element of the landscape of cities such as Stockholm, Copenhagen and Lubeck. The transformations of the late 1980s and early 1990s, "the autumn of peoples" in 1989 and the collapse of the USSR in 1991 brought further border changes in this part of Europe. There was also a reunification of Germany. After 50 years, Lithuania, Latvia and Estonia regained independence in 1991. Apart from Russia, all the countries bordering

the Baltic Sea joined the European Union, and most joined NATO. The hope was born that, after centuries of wars and conflicts, the free countries of Baltic Europe would establish close cooperation, and that the region would become an oasis of peace and prosperity. Territorial cooperation programs, including transnational programs dedicated to the Baltic Sea Region (Figure 7), provided an opportunity for the socio-economic development of the region [94].

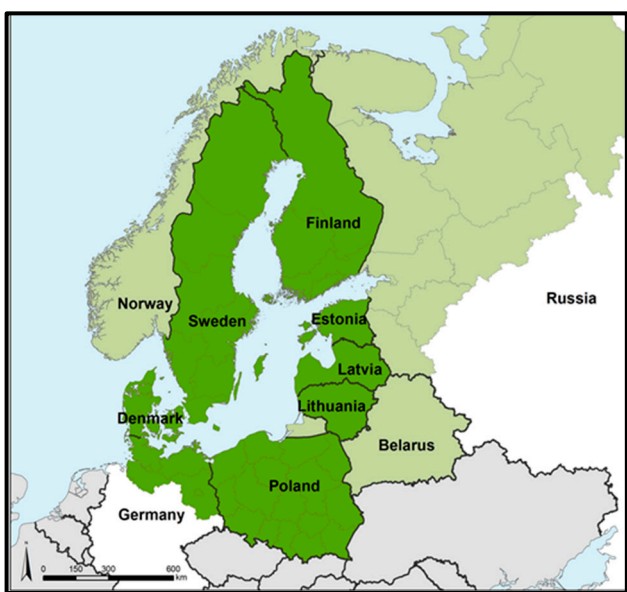

**Figure 7.** The INTERREG program in the Baltic Sea Region. Source [95].

## 4. Ethno-National Minorities in the BSR

The emergence and development of national and ethnic minorities is the result of the stormy changes that have taken place in the Baltic Sea Region in recent centuries. Describing the national and ethnic structure of individual countries is made extremely complex by differences in definitions, methods of data collection and principles of conducting censuses. In many countries, the authorities identify a limited, small number of minorities. Using official data, the percentages of national minorities (understood as those minorities that have their own states) are presented in Table 5. Jews differ in status from country to country and are classified differently. However, they are included in this Table because there is the State of Israel. However, they are also described in the next table as an ethnic minority.

**Table 5.** National minorities in BSR countries.

| No | States Nations | PL 2011 | DE 2011 | DK 2011 | SE 2010 | NO 2011 | FI 2014 | EE 2011 | LV 2011 | LT 2011 | RU 2010 | BY 2009 |
|----|------|------|------|------|------|------|------|------|------|------|------|------|
| 1. | Poles | X | 0.48 | 0.54 | 0.68 | 1.15 | 0.07 | 0.13 | 2.16 | 6.58 | 0.03 | 3.17 |
| 2. | Germans | 0.33 | x | 0.56 | 0.46 | 0.46 | 0.07 | 0.12 | 0.15 | 0.08 | 0.29 | nd |
| 3. | Danes | nd | 0.02 | x | 0.44 | 0.36 | nd | nd | nd | nd | Nd | nd |
| 4. | Swedes | nd | 0.02 | 0.27 | x | 0.66 | 0.15 | 0.03 | nd | nd | Nd | nd |
| 5. | Norwegians | nd | 0.00 | 0.29 | 0.42 | x | nd | nd | nd | nd | Nd | nd |
| 6. | Finns | nd | 0.01 | 0.07 | 1.63 | 0.12 | x | nd | nd | nd | 0.01 | nd |
| 7. | Estonians | nd | 0.01 | 0.02 | nd | nd | 0.88 | x | 0.10 | 0.01 | 0.01 | nd |
| 8. | Latvians | nd | 0.02 | 0.06 | nd | nd | nd | 0.13 | x | 0.07 | 0.01 | nd |
| 9. | Lithuanians | 0.02 | 0.03 | 0.12 | 0.07 | 0.31 | nd | 0.13 | 1.18 | x | 0.02 | 0.05 |

**Table 5.** *Cont.*

| No | States Nations | PL 2011 | DE 2011 | DK 2011 | SE 2010 | NO 2011 | FI 2014 | EE 2011 | LV 2011 | LT 2011 | RU 2010 | BY 2009 |
|---|---|---|---|---|---|---|---|---|---|---|---|---|
| 10. | Russians | 0.03 | 0.22 | 0.09 | 0.15 | 0.29 | 0.56 | 25.20 | 26.91 | 5.81 | X | 8.46 |
| 11. | Belarusians | 0.12 | 0.02 | 0.01 | nd | nd | nd | nd | 3.29 | 1.19 | 0.38 | x |
| 12. | Ukrainians | 0.13 | 0.14 | 0.12 | 0.05 | 0.04 | nd | 1.74 | 2.21 | 0.54 | 1.40 | 1.71 |
| 13. | Former Yugoslavians | nd | 0.75 | 0.87 | 1.22 | 0.60 | 0.06 | nd | nd | nd | Nd | nd |
| 14. | Jews | 0.02 | 0.11 | nd | nd | nd | nd | 0.15 | 0.31 | 0.10 | 0.11 | 0.14 |
| 15. | Italians | 0.02 | 0.61 | 0.09 | 0.07 | nd | nd | nd | nd | nd | 0.00 | nd |
| 16. | Americans | 0.03 | nd | 0.14 | nd | nd | nd | nd | nd | nd | 0.00 | nd |
| 17. | British | 0.02 | nd | 0.24 | nd | 0.26 | nd | nd | nd | nd | 0.00 | nd |
| 18. | Iraqis | nd | 0.09 | 0.53 | 1.17 | 0.43 | 0.12 | nd | nd | nd | Nd | nd |
| 19. | Iranians | nd | 0.05 | 0.28 | 0.60 | 0.28 | nd | nd | nd | nd | 0.00 | nd |
| 20. | Armenians | 0.01 | 0.01 | 0.01 | nd | nd | nd | nd | nd | 0.04 | 0.00 | nd |
| 21. | Turks | nd | 1.88 | 1.08 | 0.41 | nd | nd | nd | nd | nd | 0.08 | nd |
| 22. | Other | 3.19 | 3.23 | 4.73 | 5.97 | 5.15 | 2.10 | 2.65 | 1.62 | 1.42 | 16.76 | 0.77 |
| | **Total** | 3.92 | 7.70 | 10.12 | 13.34 | 10.11 | 4.01 | 30.28 | 37.93 | 15.84 | 19.10 | 14.30 |

Source: own study based on [96–106].

Based on our own research, with the use of information collected from individual BSR countries, 18 ethnic minorities were identified as communities that do not have their own country (Table 6).

**Table 6.** Caption title. Ethnic groups in the Baltic Sea Region.

| No. | Ethnic Group | Characteristics of Ethnic Group |
|---|---|---|
| 1. | Kashubians | Ethnic group of Slavic origin. Lives in the northern parts of Poland (Pomorskie and Zachodniopomorskie voivodeships). There are approximately 216,000 Kashubians living in Poland. They use the Kashubian language, which is a Slavic language. The Kashubian language is an official regional language in Poland. The vast majority of Kashubians are Roman Catholic. |
| 2. | Silesians | Ethnic group of Slavic origin. Lives in southern Poland (Śląskie, Opolskie, Dolnośląskie voivodeships) and the northern part of the Czech Republic and Slovakia. About 847,000 Silesians live in Poland. The Polish authorities do not recognize Silesians as a national or ethnic group. The Silesian language is recognised in Poland as a dialect of the Polish language. Silesians are Roman Catholic or Protestant (of which, mainly Lutheran). |
| 3. | Roma | Non-territorial nation or ethnic group of Indian origin whose members form a diaspora inhabiting most countries of the world, including all countries of the Baltic Sea Region. The Roma community constitutes about 10–12 million people. About 300–400,000 Roma live in BSR countries (most of whom live in Germany). The Roma language is the only Indo-Aryan indigenous language spoken in Europe. In many countries (e.g., Poland), the Roma are the most evenly territorially distributed of the national and ethnic minorities. |
| 4. | Lemkos | Ethnic group of Slavic origin, forming part of the Ruthenian nation. Originally lived in the Beskidy Mountains (the area of today's Poland–Ukraine–Slovakia borderland). Resettlement campaigns resulted in them mainly inhabiting western Poland (Dolnośląskie Voivodeship). The Lemko community in Poland is about 10,000 people strong. The Lemko language is an East Slavic language and similar to Ukrainian. Lemkos are primarily Orthodox or Greek Catholic. |
| 5. | Karaims | Ethnic and religious group of around 2000 people of Turkish or Semitic origin. Karaims live in small communities in Lithuania (Vilnius area), Poland (Warsaw or Wrocław area), as well as Russia and Ukraine. Karaim is a Turkic language. The Karaim religion derives from Judaism. |

**Table 6.** *Cont.*

| No. | Ethnic Group | Characteristics of Ethnic Group |
|---|---|---|
| 6. | Tatars | A group of peoples of Turkish origin. The community is about 6 million strong. Most Tatars (over 5 million) live in Russia, primarily in the Republic of Tatarstan. A Tatar minority is found in each BSR country and totals (excluding Russia) about 6000–8000 people. The Tatar language is a Turkic language. The Tatars mainly profess Islam. |
| 7. | Livonians | The ethnic group of Balto-Finnish origin has lived in Latvia, Courland (about 200 people) and Estonia (22 people). The Livonian language, related to Estonian, is a dying language belonging to the Finno-Ugric language group. Livonians are Lutheran. |
| 8. | Suiti | Ethnic group of Balto-Finnish origin that lives in Latvia (Courland, Alsunga commune). The community has about 300 people. The Suiti language is a Baltic language. The Suiti are Catholic. |
| 9. | Setos | Ethnic group of Balto-Finnish origin that lives on the border of Estonia (Setomaa) and Russia (Pskov Oblast). There are approximately 15,000 Setos in the world, most of them in Estonia. Seto is a Finno-Ugric language. The Setos are Orthodox. |
| 10. | Sami | Indigenous ethnic group of Georgian origin living in the Sápmi region (formerly known as Lapland), which today covers large parts of northern Norway, Sweden, Finland and Russia (Murmansk Oblast). The Sami community is of about 80,000 people (50,000 in Norway, 20,000 in Sweden, 8000 in Finland, 2000 in Russia). The languages spoken by the Sami belong to the Finno-Ugric group of languages. The Sami profess Christianity (Lutheranism, Orthodoxy) and the indigenous animistic religion. |
| 11. | Karelians | Ethnic group of Balto-Finnish origin that inhabited the historic region of Karelia that is today divided between Finland and Russia. The Karelian community numbers about 70,000 people, of whom over 60,000 live in Russia. The Karelian language is a Finno-Ugric language. The Karelians profess Christianity (Orthodoxy, Lutheranism) |
| 12. | Kvens | Ethnic group of Balto-Finnish origin that inhabited the historic region of Kvenland off the coast of the Gulf of Bothnia. The Kven community is of about 10–15,000 people and lives in northern Norway. The Kven language is a Finno-Ugric language. The Kvens profess Lutheranism. |
| 13. | Tornedalians | An ethnic group of Finnish origin that lives in northern Sweden, in the Torne region, which is divided between Sweden and Finland. The Tornedalian community has about 50–75 thousand. people. The Meänkieli language spoken by the Tornedalians is related to the Finnish language. The Tornedalians profess Lutheranism. |
| 14. | Faroese | Scandinavian nation inhabiting the Faroe Islands (58,000 people). About 20,000 people live in Denmark and 1500 in Norway. Faroese belongs to the Scandinavian group of Germanic languages. |
| 15. | Inuits | Indigenous ethnic group (also called Greenlanders) living in Greenland (56,000 people). About 10,000 people live in Denmark and 300 persons in Norway. The Greenlandic language belongs to the Eskimo-Aleutian family. Greenlanders profess Christianity (Lutheranism) and the indigenous animist religion. |
| 16. | Frisians | Indigenous ethnic minority of Germanic origin living in the provinces of Groningen and Friesland of the Netherlands, in the federal states of Lower Saxony and Schleswig-Holstein of Germany and in Denmark (South Jutland). The Frisian community has about 500,000 members, of which about 60,000 live in Germany. The Frisian language, which has many dialects, belongs to the West Germanic group. Most Frisians are Protestant (Lutheranism and Calvinism) |
| 17. | Sorbs | National–ethnic group of Slavic origin living in Germany (the federal states of Saxony and Brandenburg), and numbering around 60,000 people. Sorbs are Roman Catholic and Lutheran. The Sorbian languages (Lower Sorbian and Upper Sorbian) belong to the group of West Slavic languages. |
| 18. | Jews | Semitic nation living in Palestine in antiquity and using the Hebrew language at that time. The Jewish community numbers about 15 million people. About 170,000 people live in the Baltic Sea Region (apart from Russia, where the Jewish community numbers about 150,000). 118 000 Jews live in Germany. Hebrew, which belongs to the group of Canaanite Semitic languages, is spoken by a total of 9 million people. |

Source: own study based on [107–127].

National majorities constituted between 62.07% of the national population in Latvia and 96.38% in Poland (Figure 8).

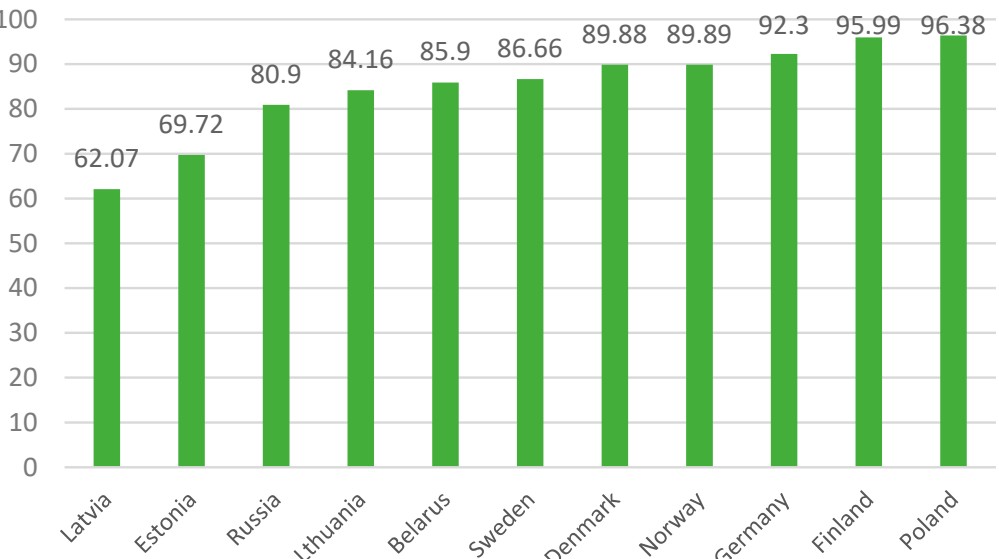

**Figure 8.** Percentage share of national majorities in BSR countries. Source: own study.

It is noteworthy that there were countries in the BSR where the societal majority was growing relative to the share of minorities (Belarus) and countries such as Sweden where the opposite trend was found (Figure 9).

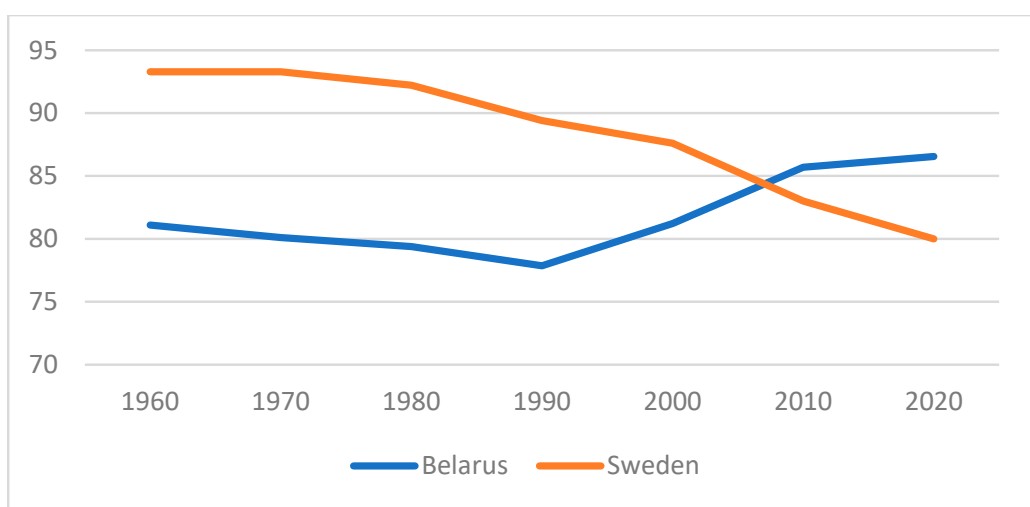

**Figure 9.** Percentage shares of the Belarusian majority in Belarus and the Swedish majority in Sweden in the years 1960–2020. Source: own study based on [106,128].

The largest minority (a Russian minority) was found in Latvia and Estonia, and the smallest minority (a German minority) in Poland (Figure 10).

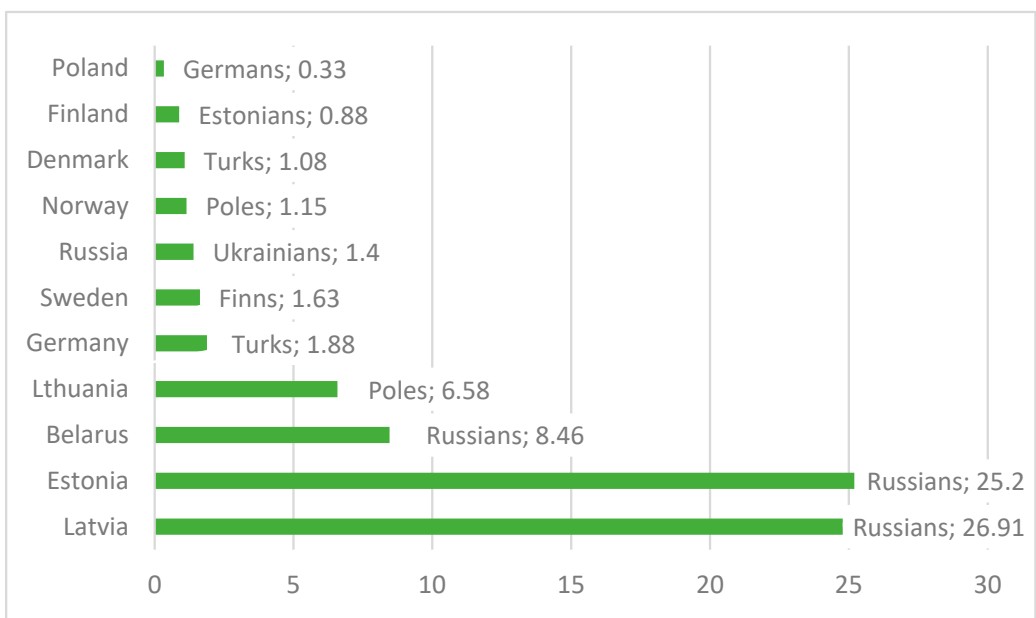

**Figure 10.** Percentage shares of the most numerous national minorities in BSR countries. Source: own study.

The largest official numbers of ethnic minorities were recorded in Poland and Sweden, and the smallest in Belarus (Figure 11).

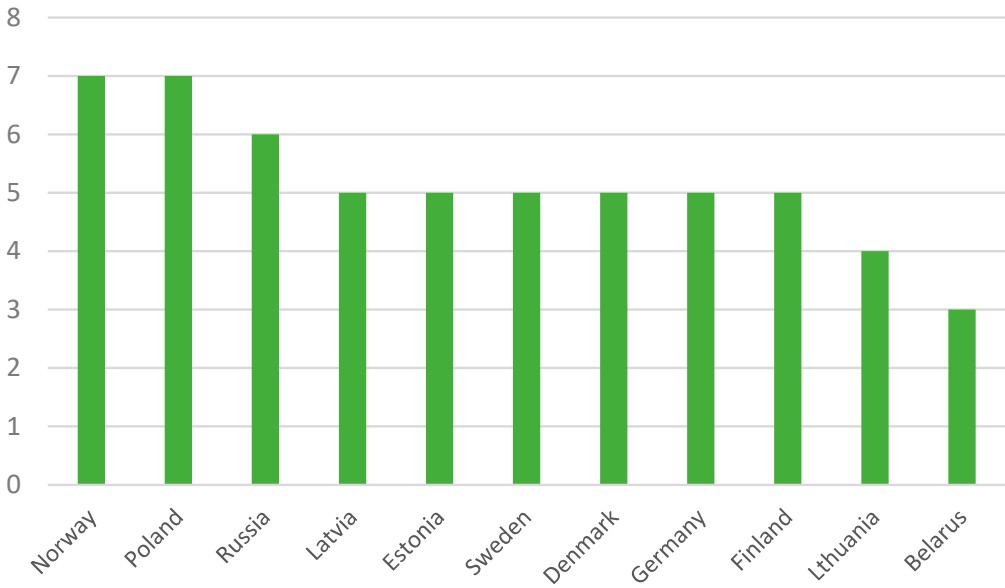

**Figure 11.** Numbers of major ethnic minority groups in the BSR countries. Source: own study.

Fewer than half of the minority groups were limited to only one country (Figure 12).

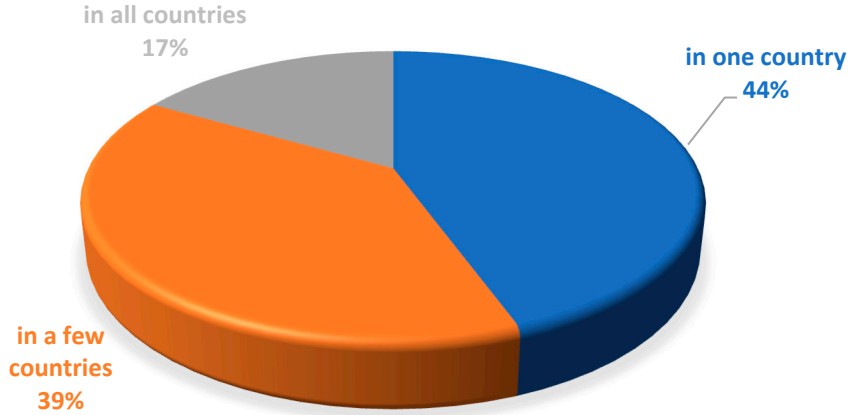

**Figure 12.** Division of ethnic minority groups in BSR countries according to the number of countries in which they occur. Source: own study.

## 5. Quantitative Analysis of the Impact of Territorial Cooperation on the Development of National and Ethnic Minorities

To analyse the impact of territorial cooperation on the development of national and ethnic minorities, a project selection concept was developed. The first stage of selection included the identification of all projects whose descriptions contained proper names of national and ethnic minorities or words such as "minority" or "ethnic" (Figure 13A). In the next stage, each of the projects was verified. Ultimately, 126 projects were selected (Figure 13B).

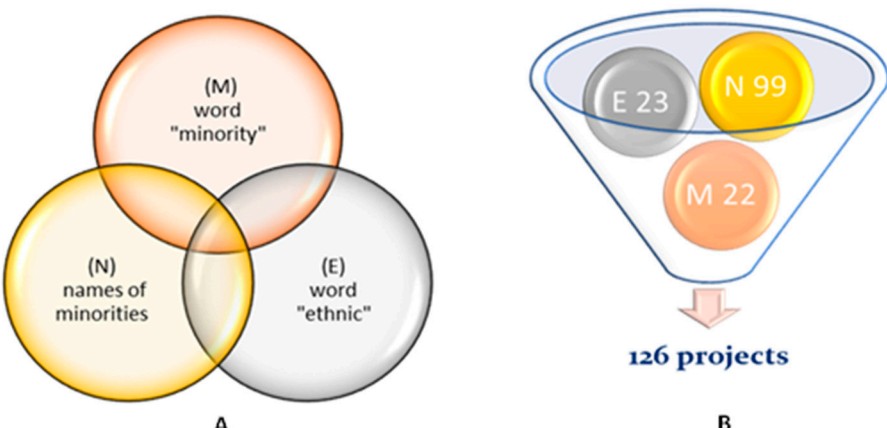

**Figure 13.** Concept (**A**) and result (**B**) of the selection process of territorial cooperation projects. Source: own study.

The numbers of projects and sizes of budgets implemented under EU programs are presented in Table 7.

**Table 7.** Numbers of projects and sizes of budgets implemented under EU programs.

| No. | Programme | Numbe of Projects | Project Budget |
|---|---|---|---|
| 1. | Programme 2000-06 Baltic Sea Region | 2 | 4,227,722 |
| 2. | Programme 2000-06 Cadses | 1 | 1,850,000 |
| 3. | Programme 2000-06 Estonia–Latvia–Russia (EE-LV-RU) | 1 | 290,700 |
| 4. | Programme 2000-06 Euregio–Karelia (FI-RU) | 1 | 39,300 |

**Table 7.** *Cont.*

| No. | Programme | Numbe of Projects | Project Budget |
|---|---|---|---|
| 5. | Programme 2000-06 Interreg IIIC North | 1 | 1,014,766 |
| 6. | Programme 2000-06 Interreg IIIC West | 1 | 562,826 |
| 7. | Programme 2000-06 Kvarken–Mittskandia (FI-SE-NO) | 1 | 3,360,000 |
| 8. | Programme 2000-06 Lithuania–Poland–Russia (LT-PL-RU) | 3 | 1,486,453 |
| 9. | Programme 2000-06 Northern Periphery | 1 | 1,001,120 |
| 10. | Programme 2000-06 Sweden–Norway (SE-NO) | 6 | 785,927 |
| 11. | Programme 2000-06 URBACT I | 2 | 1,918,529 |
| 12. | Programme 2007-13 Central Baltic (FI-SE-EE-LA) | 2 | 1,707,239 |
| 13. | Programme 2007-13 Central Europe | 1 | 1,176,810 |
| 14. | Programme 2007-13 Czech Republic–Poland (CZ-PL) | 1 | 236,134 |
| 15. | Programme 2007-13 Estonia–Latvia (EE-LV) | 1 | 126,247 |
| 16. | Programme 2007-13 Interreg IVC | 1 | 1,695,884 |
| 17. | Programme 2007-13 Kolarctic ENPI CBC | 4 | 3,678,052 |
| 18. | Programme 2007-13 Latvia-Lithuania (LV-LT) | 1 | 179,994 |
| 19. | Programme 2007-13 Latvia-Lithuania-Belarus ENPI CBC | 1 | 296,785 |
| 20. | Programme 2007-13 Nord (SE-FI-NO) | 30 | 11,318,307 |
| 21. | Programme 2007-13 Northern Periphery | 1 | 1,312,955 |
| 22. | Programme 2007-13 Poland–Slovak Republic (PL-SK) | 3 | 1,380,989 |
| 23. | Programme 2007-13 Poland-Belarus-Ukraine ENPI CBC | 2 | 1,728,346 |
| 24. | Programme 2007-13 Sweden–Norway (SE-NO) | 1 | 12,222 |
| 25. | Programme 2007-13 Syddanmark-Schleswig-K.E.R.N. | 2 | 1,201,404 |
| 26. | Programme 2014-20 Estonia–Russia ENI CBC | 4 | 1,232,809 |
| 27. | Programme 2014-20 INTERREG V-A Czech Republic–Poland | 4 | 9,628,154 |
| 28. | Programme 2014-20 INTERREG V-A Estonia–Latvia | 2 | 2,093,119 |
| 29. | Programme 2014-20 INTERREG V-A Finland–Estonia–Latvia–Sweden (Central Baltic) | 4 | 2,772,060 |
| 30. | Programme 2014-20 INTERREG V-A Germany–Denmark | 1 | 109,555 |
| 31. | Programme 2014-20 INTERREG V-A Lithuania–Poland | 2 | 996,635 |
| 32. | Programme 2014-20 INTERREG V-A Poland–Denmark–Germany–Lithuania–Sweden (South Baltic) | 1 | 2,697,116 |
| 33. | Programme 2014-20 INTERREG V-A Sweden–Finland–Norway (Nord) | 26 | 14,424,929 |
| 34. | Programme 2014-20 INTERREG VB Northern Periphery and Arctic | 5 | 4,647,438 |
| 35. | Programme 2014-20 Karelia ENI CBC | 1 | 46,311 |
| 36. | Programme 2014-20 Kolarctic ENI CBC | 1 | 1,084,909 |
| 37. | Programme 2014-20 Latvia–Lithuania–Belarus ENI CBC | 1 | 2,777,778 |
| 38. | Programme 2014-20 Poland–Belarus–Ukraine ENI CBC | 3 | 149,250 |
| | Together | 126 | 85,248,774 |

Source [129].

The number of projects and their value in subsequent EU financial periods showed an upward trend (Figure 14).

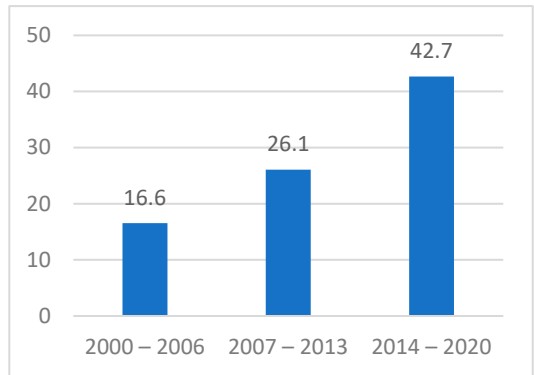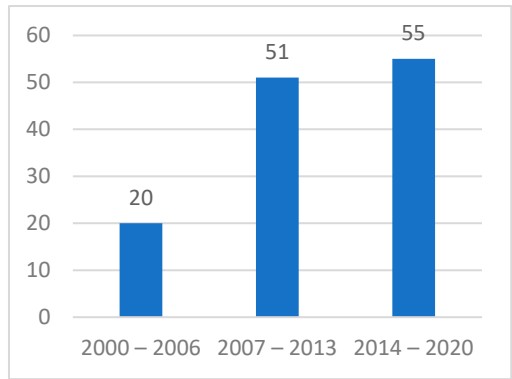

**Figure 14.** Number and value of projects (EUR million) in individual EU programming periods. Source: own study based on Keep.eu.

When analyzing the structure of the projects, it can be noticed that most of the projects (62, 11%) were in the "heritage preservation" category. The next categorie s were "social support" (24%), "education" (10.3%) and "political empowerment" (3.57%). Most funds were allocated to projects in the: heritage preservation" category (59.52%), and the least for projects in the "political empowerment" category (3.17%) (Figure 15).

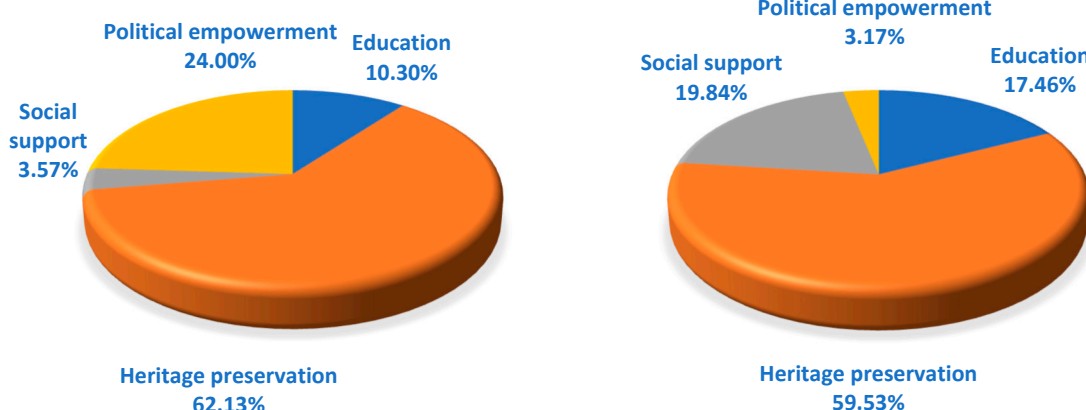

**Figure 15.** Percentage of individual types of projects by number of projects (**left**) and budget of the projects (**right**). Source: own study based on Keep.eu.

The number and structure of projects supporting ethnic minorities are presented in Table 8.

**Table 8.** Number and structure of projects supporting ethnic minorities.

| No | Minority | Category of Projects | | | | Total |
|---|---|---|---|---|---|---|
| | | Heritage Preservation | Social Support | Education | Political Empowerment | |
| 1. | Kashubians | 2 | | | | 2 |
| 2. | Silesians | 5 | | | | 5 |
| 3. | Roma | 1 | 3 | | | 4 |
| 4. | Lemkos | 2 | | | | 2 |
| 5. | Tatars | 1 | | | | 1 |
| 6. | Livonians | 1 | | | | 1 |
| 7. | Suiti | 1 | | 1 | | 2 |

**Table 8.** *Cont.*

| No | Minority | Category of Projects | | | | Total |
|---|---|---|---|---|---|---|
| | | Heritage Preservation | Social Support | Education | Political Empowerment | |
| 8. | Setos | 3 | | 2 | | 5 |
| 9. | Sami | 39 | 17 | 11 | 4 | 71 |
| 10. | Karelians | 2 | | | | 2 |
| 11. | Tornedalians | 3 | 1 | | | 4 |
| 12. | Faroese | 1 | | | | 1 |
| 13. | Jews | 5 | | | | 5 |

Source: own study based on Keep.eu.

## 6. Qualitative Analysis of the Impact of Territorial Cooperation on the Development of National and Ethnic Minorities

### 6.1. Kashubians

Two projects were carried out. Both of them were in the "heritage preservation" category. The partners were institutions promoting the Kashubian culture. The Kashubian People's University joined the Destlink network, whose aim was to "promote the competitiveness of rural regions in the sustainable tourism development sector" [130]. Kashubian manors were included in the tourist offer of the BSR. The Museum of Kashubian-Pomeranian Literature and Music (Figure 16) located in Wejherowo became highly recognized cultural institution with a unique program of exhibitions and education on Kashubian history.

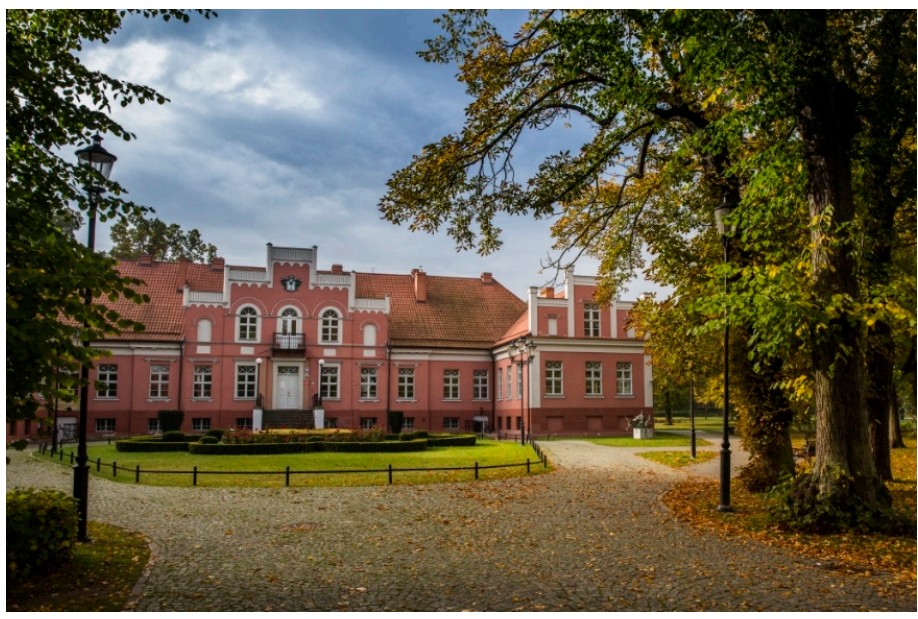

**Figure 16.** Kashubian manor house in Wejherowo (Poland). Source: [131].

### 6.2. Silesians

Four projects were carried out. All of them were in the "heritage preservation" category. They were attended by institutions developing Silesian culture and promoting tourism, including industrial tourism, which is characteristic of this region. In Opole (PL), the non-governmental organization "Silesiana" organized events promoting Silesian dance, and in Cieszyn (PL) the Folklore Center of Cieszyn Silesia was established. The Center organized cultural events in the Silesian language. (Figure 17).

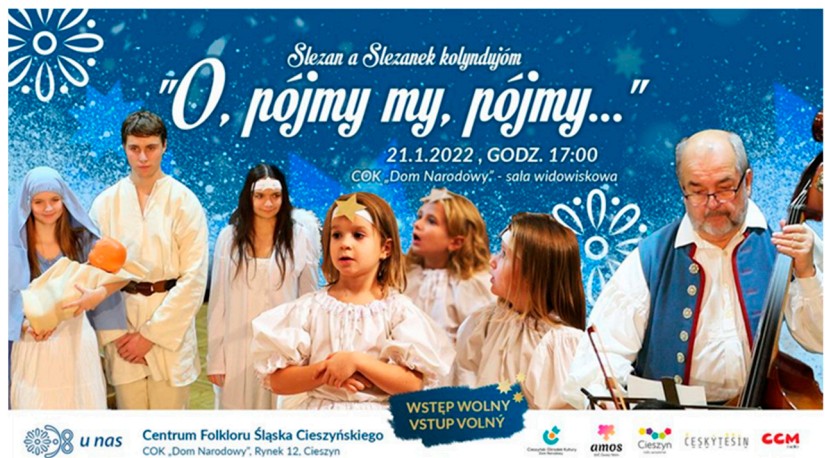

**Figure 17.** Center of Folklore of Cieszyn Silesia. source: [132].

### 6.3. Roma

Four projects were carried out, including three in social support category. Their aim was to support the process of integration of the Roma communities in their place of residence (the Finnish–Russian borderland), to help prepare community members for professional work (in Latvia and Germany), including work in the field of collecting and distributing herbs. Moreover, one of the projects ("Meeting of Seven Polish–Slovak Borderland Cultures") contributed to the promotion of Roma culture.

### 6.4. Lemko

Two projects were carried. Both of them were in the heritage preservation category. They contributed to the promotion of the Lemko culture and religion. In the village of Komańcza (PL), a Lemko church was reconstructed, which has become a cultural tourism attraction [133].

### 6.5. Tatars

One project was carried out in the heritage preservation category. As part of the project "Preservation and Promotion of Cultural Heritage of Kalvarija and Suchowola" in Poland and in Lithuania, cultural centers promoting Tatar culture were opened. The Center of Three Cultures (Polish, Tatar and Jewish cultures) in Suchowola (PL) has established close cooperation with the Tatar Culture Center of Islam, located in the same town, and jointly organized several concerts promoting Tatar culture (Figure 18).

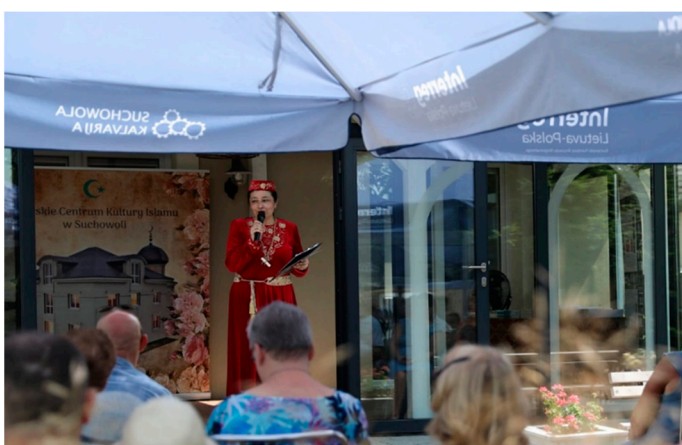

**Figure 18.** Event promoting Tatar heritage organised by the Center of Three Cultures in Suchowola (PL). Adapted with permission from Ref. [134]. 2020, Kurier Poranny.

### 6.6. Livonians

One project was carried out ("Design and Promotion of a Tourism Product based on Livonian Culinary Heritage"). This was in the heritage preservation category and promoted the Livonian culinary heritage in the Latvian–Estonian border area (Figure 19).

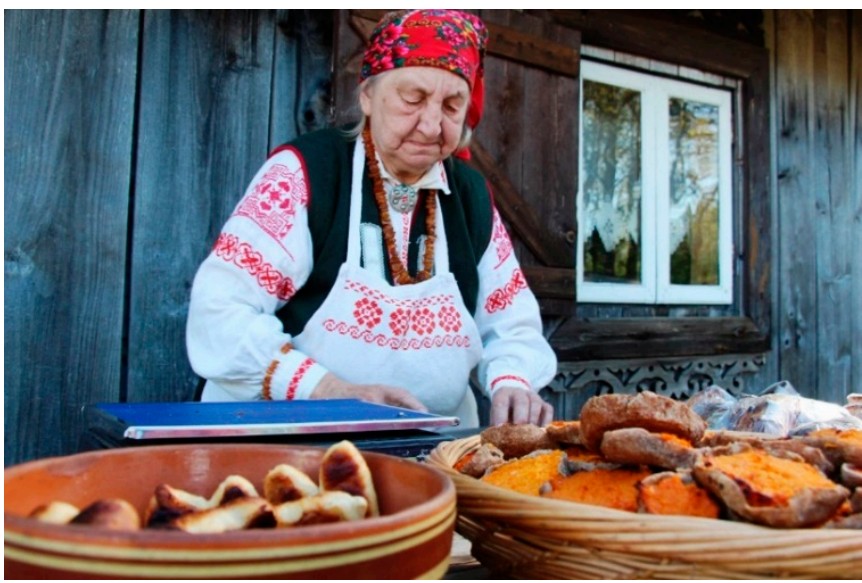

**Figure 19.** Design and promotion of tourism product based on Livonian culinary heritage. Adapted with permission from Ref. [135]. 2020, Interreg Estonia-Latvia.

### 6.7. Seto and Suiti

Five projects were carried out, including two projects related to both Suiti and Setos. One—"Renaissance of Seto and Suiti Ethnic Cultures"—concentrated on the Latvian–Estonian border. The second was to preserve and promote the heritage by preparing an offer of cultural tourism. The remaining projects in the Heritage Preservation category involved the promotion of Seto heritage, the organization of publishing activities, the organization of cultural events and the development of tourism. The educational project "Improving Cooperation between Local Authorities, Schools and NGOs in Teaching and Promotion of Local Cultural Heritage to Children and Youngsters in the Historical Setomaa Area" implemented along the Russian–Estonian border was of great importance.

### 6.8. Sami

Projects related to the Sami community constituted the most numerous group (71 projects). This was the only group in which there were projects assigned to all four categories. Projects in the heritage preservation category supported, *inter alia*, the protection of intellectual rights, scientific and research activities, the organization of music, theatre and folklore events, the modernization of museum facilities and the promotion of cultural tourism. Social support concerned professional development and social integration. Educational projects concerned, *inter alia*, history, language, climate adaptation, ecology and agricultural activities, including reindeer breeding. Projects in the political empowerment category, which appeared only in the Sami community, were of particular importance. The project was aimed at building strong relations between the institutions representing the Sami and EU structures in order to have a real influence on shaping European policy. The Sami Council (Figure 20), which represents the Sami communities of four countries, played a key role in this project.

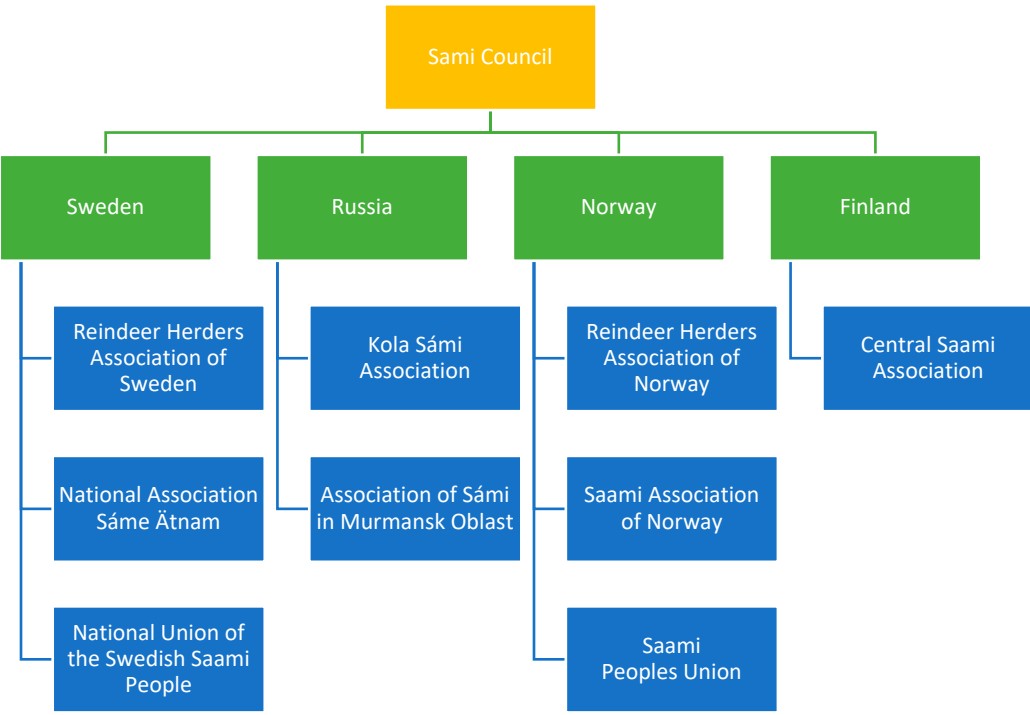

**Figure 20.** Sami Council structure. Source: own study based on [136].

The project "Strategy and Action Plan for the Barents Region up to 2010" enabled the Samis to participate in strategic planning, while the project "Sami Language—Three Generations Tell" politically renewed the status of the Sami language as an ethnic minority language.

*6.9. Karelians*

Two projects were carried. All of them were the heritage preservation category and promoted cultural tourism along the Finnish–Russian border.

*6.10. Tornedalians*

Four projects were carried out, including three projects in the heritage preservation category, supporting the cooperation of cultural institutions and the development of sustainable tourism. The collaboration platform for minority languages in the North Calotte was established and played an important role in learning and popularizing the Meänkieli language used by the Tornedalians.

*6.11. Faroes*

A project in the heritage preservation category was carried out (Destination Viking—Saga Lands), which promoted the heritage of the Sami people through the development of a cultural tourism offer.

*6.12. Jews*

The aim of all projects (5 projects in the heritage preservation category) was to strengthen the cooperation of cultural institutions and support the research and development of cultural tourism, including through the creation of Shtetl routes (Figure 21).

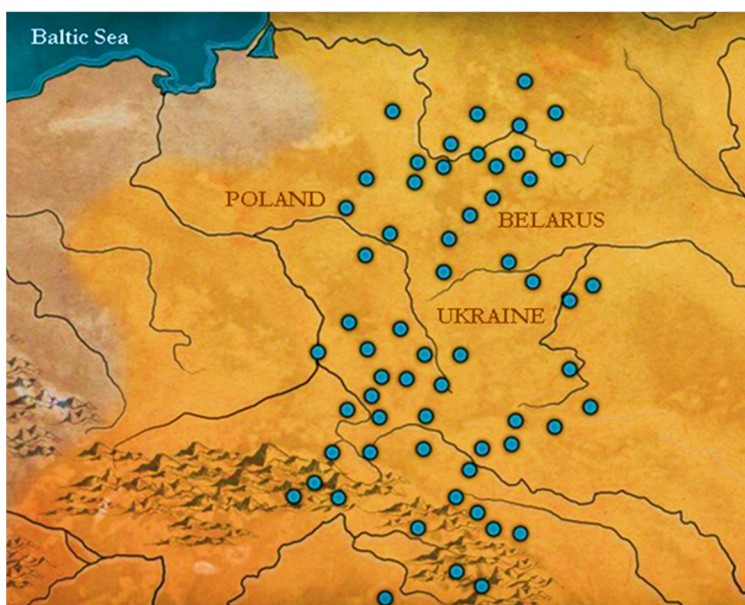

**Figure 21.** Shtetl routes promoting Jewish heritage. Source: [137].

## 7. Conclusions

For many centuries, the Baltic Sea region was an arena of confrontation between states and nations. State borders have changed many times. States have risen and fallen, and ethnic groups have fought for their rights. The democratic changes that took place at the end of the 20th century and the enlargement of the European Union to include Poland and the Baltic republics created a good climate for cooperation and an opportunity that national and ethnic minorities could take advantage of. Such is the nature of territorial cooperation programs that obtaining funds requires the acquisition of partners—and foreign partners at that.

For the period of 2000-14, 126 projects were identified. This is a relatively small number. The projects were "soft" in nature. The project budgets, which oscillated around the EUR 1 million mark, did not allow for major investments. Most of the projects supported the cultural heritage of national and ethnic minorities. This heritage was promoted as a tourist attraction. Mainly minorities located in border areas took part in the projects. Projects providing social support were important. The biggest beneficiary was the Sami community, which was the only one to implement projects aimed at political empowerment. The project establishing the Sami Council deserves special attention.

Neighborhood and EU cohesion programs have become effective tools to support the development of minorities. It seems that the potential of national and ethnic organizations has not been exploited; perhaps they lacked the strength and ability to raise external funds. It is worth publicizing this issue more as part of the European Union Strategy for the Baltic Sea Region. Key decision-makers from the region have the ambition to make Baltic Europe a leader in sustainable development, and this goal requires partner cooperation with minoritie, including ethnic and national minorities.

**Author Contributions:** Conceptualization, T.S.; methodology, T.S. and B.M.; software, T.S. and B.M.; validation, T.S. and B.M.; formal analysis, T.S. and B.M.; investigation, T.S.; resources, T.S. and B.M.; data curation, T.S.; writing—original draft preparation, T.S.; writing—review and editing, T.S. and B.M.; visualization, T.S.; supervision, B.M.; funding acquisition, T.S. and B.M. All authors have read and agreed to the published version of the manuscript.

**Funding:** The project is financed within the framework of the program of the Minister of Science and Higher Education under the "Regional Excellence Initiative" in the years 2019–2022, project number 001/RID/2018/19, with the amount of financing being PLN 10,684,000.00. This research was also funded by Gdynia Maritime University fund WZNJ/2022/PZ/09.

**Institutional Review Board Statement:** Not applicable for studies not involving humans or animals.

**Informed Consent Statement:** Not applicable for studies not involving humans.

**Data Availability Statement:** Not applicable.

**Conflicts of Interest:** The authors declare no conflict of interest.

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
