# Peer review of "The Use of EU Territorial Cooperation Funds for the Sustainable Development of National and Ethnic Minorities in the Baltic Sea Region"

_sustainability, doi:10.3390/su14137729_

Round 1

Reviewer 1 Report

The authors choose very interesting topic and it is not very well investigated.

The aim of the research and methodology is not described in the Introduction.

I am not sure, that the authors correctly showed the source of the Figure 1. What does it means here: Source: own study based on?

In my opinion data in Table 4 is not very good visible. Better to format it on the left side of the table.

The authors just provide a figure of the model (see Figure 6), but did not describe it at all. It is necessary to describe the elements, but the most important to provide a description how the model could be used.

In Figure 12 not all names of countries are clear visible.

It is not understandable what is represented in Figure 16. What is the different between two charts? Please, provide the description.

In 6 sub-chapter the authors provide the description of the projects in different ethnic minorities. But the title of the sub-chapter is "Qualitative analysis of the impact of territorial cooperation on the development of national and ethnic minorities"

However, the analysis is not provided, the only description of the projects. Also, an impact of the projects is not described. The authors have to provide the description to show reader why they described all of these projects.

References are not formatted according to the requirements.

Author Response

Dear Reviewer,
We would like to thank you for your effort and time spent on the revision of our paper. This is not just a routine “thank you.” We put a lot of energy into carrying out time-consuming research and preparing the text. And now we are grateful that thanks to your suggestions the paper looks better.  We have modified the structure of the article, improved the text, and did our best to eliminate our mistakes.

Below we present responses to your valuable suggestions:

1.    The aim of the research and methodology is not described in the Introduction. – We agree and corrected it.
2.    I am not sure, that the authors correctly showed the source of the Figure 1. What does it means here: Source: own study based on? . – We agree and corrected it.
3.    In my opinion data in Table 4 is not very good visible. Better to format it on the left side of the table. – We agree and corrected it.
4.    The authors just provide a figure of the model (see Figure 6), but did not describe it at all. It is necessary to describe the elements, but the most important to provide a description how the model could be used. We agree. Corrected it, the description has been added.
5.    In Figure 12 not all names of countries are clear visible.    – We agree and corrected it.
6.     It is not understandable what is represented in Figure 16. What is the different between two charts? Please, provide the description.     We agree and corrected it.
7.    In 6 sub-chapter the authors provide the description of the projects in different ethnic minorities. But the title of the sub-chapter is "Qualitative analysis of the impact of territorial cooperation on the development of national and ethnic minorities" However, the analysis is not provided, the only description of the projects. Also, the impact of the projects is not described. The authors have to provide the description to show reader why they described all of these projects. We partly agree. You are right that the title "Qualitative analysis of the impact of territorial cooperation on the development of national and ethnic minorities" might have sounded too ambitious. In this research, however, we were not able to describe more than 100 projects in detail. Undoubtedly, it can be done in the future. The article is now quite long. That's why we tried to find a compromise. The chapter title has been modified and the introduction has been added. A new photo has been inserted that well illustrates the support of an ethnic minority by INTERREG funds.

Regards,
The authors

Reviewer 2 Report

Review on the paper entitled “The use of EU territorial cooperation funds for the sustainable development of national and ethnic minorities in the Baltic Sea Region”.

This is a nice paper about the relationship between sustainability and minority issues in the Baltic Sea Region. The topic being investigated by the authors is very interesting and, on the other hand, very unusual, primarily in the context of sustainability. I think that the authors made a huge effort to demonstrate how the EU cohesion funds affected the development of ethnic minorities in the region.

My overall opinion about the paper is that it is of high quality including the conceptualization of the research, the figures, and tables. However, I recommend the authors to implement some modifications in the paper.

The theoretical sections dealing with the conceptualization of sustainability and minorities are impressive and very nicely presented by the authors. However, a regular Introduction section is missing from the study. The Introduction is currently focusing on describing the concept of sustainability. This is not what readers expect to be found in the Introduction. In that section, the authors should briefly demonstrate their motivation to conduct the research, the geographical, historical, socio-economic context of the research, the importance of the research, the organization of the paper, and finally, they must set the research questions. These components are completely missing from the Introduction, but we find there a nice description of sustainability. In addition, it is unusual to put figures in the Introduction.

So, I recommend the authors, to write a straightforward Introduction for the paper, that precedes the literature review section, and/or the theoretical background section (etc.).

Finally, in the Conclusions section, this can be read (lines 528-529): “Neighbourhood and EU cohesion programmes have become an interesting tool to support the development of minorities.” I think the word “interesting” is not suitable in this context.

Author Response

Dear Reviewer,

We would like to thank you for your effort and time spent on the revision of our paper. This is not just a routine “thank you.” We put a lot of energy into carrying out time-consuming research and preparing the text. And now we are grateful that thanks to your suggestions the paper looks better.  We have modified the structure of the article, improved the text, and did our best to eliminate our mistakes.

Below we present responses to your valuable suggestions:
1.     Introduction section is missing from the study. The Introduction is currently focusing on describing the concept of sustainability. This is not what readers expect to be found in the Introduction. In that section, the authors should briefly demonstrate their motivation to conduct the research, the geographical, historical, socio-economic context of the research, the importance of the research, the organization of the paper, and finally, they must set the research questions. These components are completely missing from the Introduction, but we find there a nice description of sustainability. In addition, it is unusual to put figures in the Introduction. So, I recommend the authors, to write a straightforward Introduction for the paper, that precedes the literature review section, and/or the theoretical background section (etc.).

Yes, you are right. The paper missed a classical introduction. We change the structure of the paper. Prepared a new introduction and modified the next chapter.
2.    In addition, it is unusual to put figures in the Introduction.  We agree and corrected it.
3.    in the Conclusions section, this can be read (lines 528-529): “Neighbourhood and EU cohesion programmes have become an interesting tool to support the development of minorities.” I think the word “interesting” is not suitable in this context. We agree and corrected it.
Regards,
The authors

Round 2

Reviewer 1 Report

The authors improved their paper according to the reviewer' comments. However, there are still some improvements required:

Line 282. Environment is mentioned twice, also in brackets. Why?

Line 448. Still it is not understandable what is represented in Figure 16. The description is not provided.

Check if references are formatted according to the requirements, especially web sites. It seems that it is not full description provided.

Author Response

Dear Reviewer,

I agree with your suggestions and believe I have made all corrections.

Regards

Tomasz Studzieniecki